A phylogeographic pattern of the trans-Palaearctic littoral water flea Pleuroхus truncatus (O.F. Müller, 1785) (Cladocera: Chydoridae)

http://orcid.org/0000-0002-8863-6438 Kotov Alexey A. 1 alexey-a-kotov@yandex.ru
Garibian Petr G. 1
Neretina Anna N. 1
http://orcid.org/0000-0001-6008-7441 Karabanov Dmitry P. 1 2
1 A. N. Severtsov Institute of Ecology and Evolution of Russian Academy of Sciences , Moscow , Russia
2 Papanin Institute for Biology of Inland Waters of Russian Academy of Sciences , Borok, Yaroslavl Area , Russia
Zhang Lin
Electronic publication date: 2025 Apr 24
Publication date: 2025
Volume: 13
Electronic Location ID: e19355
Received 2025 Jan 16; Accepted 2025 Apr 1
Copyright: © 2025 Kotov et al.
Copyright year: 2025
Copyright holder: Kotov et al.
License: This is an open access article distributed under the terms of the Creative Commons Attribution License, which permits unrestricted use, distribution, reproduction and adaptation in any medium and for any purpose provided that it is properly attributed. For attribution, the original author(s), title, publication source (PeerJ) and either DOI or URL of the article must be cited.
License URL: https://creativecommons.org/licenses/by/4.0/

Keywords: Phylogeography, Phylogeny, Molecular clock, Cladocera, Eurasia

Funding: Russian Science Foundation 23-14-00128 The study is supported by the Russian Science Foundation (Grant 23-14-00128). The funders had no role in study design, data collection and analysis, decision to publish, or preparation of the manuscript.

==============================
Water fleas (Crustacea: Cladocera) are recently regarded as models for phylogeographic studies, but most such publications are concerned the planktonic genera, first of all, Daphnia O.F. Müller. The aim of our article is to study the phylogeographic pattern of a very common littoral chydorid Pleuroxus truncatus (O.F. Müller) based on sequences of two mitochondrial (cytochrome oxidase (COI) and 16S) and two nuclear (18S and 28S) genes. All totality of the sequences could be subdivided into two major clades, A, having a predominantly European distribution with a single exclusion, and B, having a predominantly Asian distribution, but with few populations in European Russia; the clade B is subdivided into three subclades (B1–B3) with a moderate support. Earlier derived phylogroups (subclades B1 and B2) are distributed in south portion of Western Siberia. This pattern is known for previously studied daphniids. Estimations of the major clade (A and B) and subclade differentiation time in P. truncatus based on different methods lie in the interval of ca. 0.01–0.3 Mya. Therefore, the whole revealed pattern is very young, related to Late Pleistocene and more shallow as compared to previously studied daphniids. Probably, the total population of P. truncatus was not so strongly split by the unfavorable conditions during the Pleistocene cold phases.

Introduction

Cladocerans (Crustacea: Branchiopoda: Cladocera) are distributed everywhere in continental waters, and are frequently dominant among the microscopic planktonic and littoral microcrustaceans (Dumont & Negrea, 2002). Four recent orders are known: Anomopoda Sars, 1865 (very large and speciose), Ctenopoda Sars, 1865 (with ca. 70 known species), Onychopoda Sars, 1865 (with ca. 35 species) and Haplopoda Sars, 1865 (with two species only). Cladocerans are well-known models of recent phylogenetic and phylogeographic studies in continental waters, but, in reality 95% of such investigations are focused on a single genus: Daphnia O.F. Müller, 1776 (Daphniidae) (Adamowicz et al., 2009; Petrusek et al., 2009; Crease et al., 2012; Ma et al., 2015; Zuykova et al., 2018b; Zuykova et al., 2019). Other groups are studied mainly sporadically, with few exceptions such as planktonic Bosmina Baird, 1845 (Anomopoda: Bosminidae) (Faustova et al., 2010; Faustova et al., 2011; Wang et al., 2019) and Diaphanosoma Fischer, 1850 (Liu et al., 2018), or few predatory planktonic genera of Onychopoda and Haplopoda (Cristescu & Hebert, 2002; Xu et al., 2011), including those being known as invasive species with great harmful economic effect (Karpowicz et al., 2024).

Representatives of a single anomopod family, Chydoridae Dybowski et Grochowski, 1894, comprise 40% of the total cladoceran species number in Northern Eurasia (113 of 287 taxa) (Korovchinsky et al., 2021). Chydorids are also dominate in the world fauna (e.g., 269 among 620 species according to Forró et al. (2008), although such numbers are now outdated). This family is extremely diverse in morphology and life styles, exploring different niches in the littoral zone, from floating leafs to bottom sediments (Fryer, 1968; Smirnov, 1996), but only few species are able to live in the pelagic zone (Smirnov, 1971). Among chydorids, a few genera are corps-eaters and even predators (Fryer, 1968; Van Damme & Dumont, 2007) which is very unusual for anomopods. Chydorids had already differentiated in the Palaeozoic (Sacherova & Hebert, 2003) and represent one of most successful groups of littoral microscopic animals (Smirnov, 1971).

Phylogeographic patterns of the aforementioned well studied planktonic animals could be different from those of littoral taxa. It could be expected due to different methods of resting egg dispersion: the latter are easily dispersed in planktonic forms, but in littoral forms they are frequently adapted to be kept in the water body where they were produced (Fryer, 1972).

A single chydorid species group with a relatively well-studied phylogeography is Chydorus sphaericus (O.F. Müller, 1776) s. lat. (Belyaeva & Taylor, 2009; Kotov et al., 2016; Wang et al., 2021; Karabanov et al., 2022a). The general phylogeographic pattern of this group in Northern Eurasia appears basically similar to that of some Daphnia (Bekker et al., 2018; Zuykova et al., 2018a, 2019). Recently two major clusters of phylogroups can be recognized, with centers of their Late Pleistocene dispersion in (1) western and (2) eastern portions of Northern Eurasia–western portion of North America = so-called “Beringian” zone. A transitional zone (= zone of their secondary contact) is located in Eastern Siberia, in the Yenisey or Ob’ river basins (Kotov et al., 2016). Also it was demonstrated for the species groups studied in phylogeographic aspect that some relict clades have also survived in Eastern Siberia during whole period of the Pleistocene glaciation cycles (Bekker et al., 2018; Zuykova et al., 2019), but such a pattern was not discussed in Chydorus.

Note that other species of the genus Chydorus Leach, 1816 are almost untouched by phylogeographic studies; any genetic information on them is very scarce and lacking a phylogeographic interpretation (Elias-Gutierrez et al., 2008). Very preliminary phylogeographic studies concerning the genera Alonella Sars, 1862 (Neretina et al., 2021) and Biapertura Smirnov, 1971 (Sinev, Karabanov & Kotov, 2020) were published recently, but they were based on a limited number of populations, without attempts to trace their global Late Quaternary history.

We can conclude that, in general, phylogeographic patterns in chydorids are unknown. But we know from publications of “traditional” taxonomists that there are both locally distributed endemics of Northern Eurasia, especially in its easternmost portion (Kotov et al., 2011; Kotov & Sinev, 2011; Jeong, Kotov & Lee, 2014), and the taxa widely distributed in different regions, where they are common, if not dominant, in continental waterbody ecosystems (Smirnov, 1971). The latter need to be revised based on molecular methods (Korovchinsky et al., 2021) and are interesting for phylogeographic purposes.

Pleuroxus truncatus (O.F. Müller, 1785) (Chydoridae: Chydorinae) is among the most widely distributed taxa in Northern Eurasia, and it is very common in freshwater bodies of different types, from small pools to large lakes and macrophyte zone of large rivers. P. truncatus has a very characteristic morphology: a row of strong spines along whole posterior valve margin (Figs. 1A–1C) which is unique among all the chydorids, while two minor pores between two major ones on the head shield midline and elongated postabdomen with well-developed marginal teeth (Figs. 1D–1F) are typical of the genus Pleuroxus Baird, 1843.

Figure 1 Morphology of Pleuroxus truncatus, photos made by scanning electron microscope.

Lateral view (A); posterior view (B); posterior valve margin (C); head pores (D); postabdomen in lateral view (E); the same in dorsal view (F). Scale bars: 0.1 mm.

Just in its particular case, we can believe in previously published data (usually dubious in case of many other chydorids) for accurate drawing of the limits of its distribution range. The species is common in the whole of Europe from its most western portion—Iberian Peninsula (Alonso, 1996) to the most eastern portion—the Ural mountains, and from the most southern portion—Italy (Margaritora, 1985) to the most northern portion—Sweden (Lilljeborg, 1901), Norway (Sars, 1862) and northern portion of European Russia (Smirnov, 1971). Also, it is present in almost the whole Asian portion of Russia, although is not found on the Arctic islands (Vekhov, 2000) and most north-eastern territories: Kamchatka, Chukotka (Streletskaya, 1975, 2010; Trukhan et al., 2024) and Magadan Area (Novichkova & Chertoprud, 2022). It is present usually in the northern Primorsky Territory of Russia, but absent in South Korea (Jeong, Kotov & Lee, 2014) and Japan (Tanaka, 1989). Southern-most territories of its distribution range include Turkey (Guher, 2014), Kazakhstan (Ibrasheva & Smirnova, 1983), Mongolia (Alonso et al., 2019), North China (Ji et al., 2015). It is probably fully absent in North Africa (Brehm, 1954; Dumont, Laureys & Pensaert, 1979; Marrone et al., 2016; Ghaouaci et al., 2018).

The aim of our article is to study the phylogeographic pattern of P. truncatus in northern Eurasia and provide a new insight into the phylogeography of littoral Cladocera, incorporating a trans-Eurasian biogeographic reconstruction and molecular clock analysis.

Materials and Methods

Ethics statement

Field collection in public property in Russia does not require permissions. No samples were collected in any protected territories. Sampling in Mongolia was performed in the frame of the Joint Russian–Mongolian Complex Biological Expedition, curated by the Ministry of Nature, Environment and Tourism of Mongolia, and does not require especial permission. Field works did not affect endangered or protected species.

Field collection and provisory identification

Samples were collected in European Russia, Siberia, Far Eastern region of Russia, and Mongolia (Fig. 2, Table S1) by plankton nets or dip nets with a diameter of 0.20–0.4 m and a mesh size of 30–250 µm, and immediately preserved in 96% alcohol in the field. In the laboratory, they were preliminarily analysed under a binocular dissective microscope LOMO (Open Joint-Stock Company, Moscow, Russia). Specimens of Pleuroxus spp. were picked from the samples by pipettes and placed on slides (in a drop of glycerol) for accurate identification under a high-power optical microscope Olympus BX41 (Olympus Corporation, Japan) in toto. Few specimens from a sample collected in a pond in Moscow City were lyophilised, mounted on an aluminium stub, coated with gold, and examined under a scanning electron microscope (JEOL-840A).

Figure 2 Distribution of major Pleuroxus truncatus clades in northern Eurasia.

The map was created in free software (QGIS v.3.32.2). The base map was from the open domain plain map available at https://marble.kde.org/.

DNA sequencing, analysis of sequences and reconstruction of phylogeny

Genomic DNA was extracted from single adult females (Table S1) using the Wizard Genomic DNA Purification Kit (Promega Corp., Madison, WI, USA) according to the manufacturer protocol. Four markers were investigated here: the 5′-fragment of the first subunit of mitochondrial protein-coding marker cytochrome oxidase (COI) (Hebert, Ratnasingham & deWaard, 2003); the 5′-fragment the mitochondrial 12S rRNA gene (12S) (Antil et al., 2023); 5′-fragment of the nuclear small (18S) and large (28S) ribosomal subunits (von Reumont et al., 2009). For partially degraded samples we used the internal primers for a conservative portion of the COI fragment: a combination of F+iR and iF+R primers gives two PCR products of about 350 bp length each, with an overlap zone of about 70 bp allowing to form a full contig of COI. Primers for amplification are listed in Table 1. The PCR program included a pre-heating of 3 min at 94 °С; 40 cycles (initial denaturation of 30 s at 94 °С, annealing of 40 s at a specific temperature (Table 1), an extension of 80 s at 72 °С); and a final extension of 7 min at 72 °С. For COI, elongation time was twice reduced as compared to traditional protocol. PCR product was visualized in 1.5% agarose gel and purified according to soft protocol (Karabanov et al., 2022b). Each PCR product was sequenced bi-directionally on the ABI 3730 DNA Analyzer (Applied Biosystems, Waltham, MA, USA) using the ABI PRISM BigDye Terminator v.3.1 kit at the Syntol Co., Moscow.

Table 1 Genes, primers and annealing temperatures used in this study.

Gene	Primer	Sequence 5′–3′	Temp (°C)	
COX1	Pl-COI-F	CAC TTT AYT TCT TDT TTG GDA TTT G	46	
	Pl-COI-R	ACG AAA ARA TGT TGA TAW ARA ATA GG	46	
	Ple-COI-iF	CGA YTW AAT AAT TTA AGH TTY TGA C	46	
	Ple-COI-iR	ATC CHG TWC CWG CYC CTC TYT C	46	
16S	16Sch-a	GAC TGT GCA AAG GTA GCA TAA TC	52	
	16S-br3	CCG GTC TGA ACT CAG ATC ACG T	52	
18S	18Sa1	CCT AYC TGG TTG ATCCTGCCAGT	52	
	18S700R	CGC GGC TGC TGG CAC CAG AC	52	
28S	28Sd1a	CCC SCG TAA YTT AAG CAT AT	48	
	28Sd2b2	CGT ACT ATT GAA CTC TCT CTT	48	

Initial analysis of the chromatograms, formation of contigs and their subsequent editing was made with the Sanger Reads Editor in the Unipro uGENE v.50 packet (Okonechnikov, Golosova & Fursov, 2012). The authenticity of the sequences was verified by BLAST comparisons with published cladoceran sequences by ElasticBLAST (Camacho et al., 2023) in the NCBI GenBank (Sayers et al., 2021). Original sequences were deposited to the GenBank (accession numbers PQ798952–PQ799024 for COI, PQ799025–PQ799071 for 16S, PQ799072–PQ799121 for 18S and PQ799124–PQ799169 for 28S).

To reduce the influence of population structure on our results, we subdivided all of the specimens into large population groups (clades) in TaxonomR (Vernygora, Sperling & Dupuis, 2024) with a maximum weight of the tree branches and equal weights of other variables. Although this method allows to vary the weights of different parameters (branch lengths, age estimations, distributions), the conservative approach allows to separate genetic lineages accurately (Vernygora, Sperling & Dupuis, 2024).

Nucleotide diversity analysis, neutrality tests, disequilibrium pattern analysis, and statistics associated with population growth and divergence were carried out using dnaSP v.6.12 (Rozas et al., 2017). We applied the Fs test (Fu, 1997) and R2-statistics (Ramos-Onsins & Rozas, 2002) as most power tests to confirm neutrality and describe demographic processes (Ramirez-Soriano et al., 2008; Garrigan, Lewontin & Wakeley, 2010). Coalescent simulation (Hudson, 1990) also was performed in dnaSP v.6.12 under a wide range of demographic scenarios, such as population growth (or decline), population bottleneck, and population split with admixture.

Hierarchical analysis of molecular variance (AMOVA) (Excoffier, Smouse & Quattro, 1992) was performed in “Statistics” block of popART. We subdivided all genetic variability of COI (72 sequences belonging to 52 haplotypes) into three levels of structural hierarchical components: A general dataset—without hypotheses on hierarchical structure;

Major clades “A” and “B”;

Subclades within them.

FST was calculated for each of them based on 1,000 permutations, which does not require a hypothesis on normal distribution of the initial data set, in contrast to classical dispersion analysis.

Alignment of each locus sequences was performed independently in UGENE. Protein-coding COI was aligned based on muscle5 algorithm (Edgar, 2022) with translation to the aminoacids according to the invertebrate mitochondrial code. For loci of the ribosomal genes (16S, 18S, 28S rDNA) we used MAFFT v.7.526 algorithm (Katoh & Standley, 2013) with Q-INS-i strategy that takes into consideration the secondary structure. Linking sequences and their partitioning for subsequent analyses were made in SequenceMatrix v.1.9 (Vaidya, Lohman & Meier, 2011), conversion of different formats was performed in PGDSpider v.2.1 (Lischer & Excoffier, 2012). The best-fitting models of the nucleotide substitutions for each locus and for linked data were selected using ModelFinder v.1.6 (Kalyaanamoorthy et al., 2017) at the Center for Integrative Bioinformatics Vienna, Austria (http://www.iqtree.org) (Trifinopoulos et al., 2016).

Best models of nucleotide substitutions were selected based on minimal Bayesian information criterion (BIC) (Zhang, Yang & Ding, 2023): COI – TPM3u+F+I; 16S – HKY+F; 18S – JC+I; 28S – F81+F (Minh et al., 2022).

Phylogenetic reconstructions based on the maximum likelihood (ML) and Bayesian (BI) methods were made for each gene separately, for the full set of mitochondrial genes, for full set of nuclear genes, and for all unlinked genetic data.

For ML analysis we used IQ-TREE v.2.3.4 (Minh et al., 2022) with 10 K replicas of UFBoot2 (Hoang et al., 2018) was used to estimate the branch support values and SH-aLRT (Guindon et al., 2010) as a test for the tree topology. One would typically start to rely on the clade if its SH-aLRT >80% and UFboot >95% (Minh et al., 2022).

For BI analysis we used BEAST2 v.2.7.7 (Bouckaert et al., 2019). Parameters of the substitution model, speciation model according to Yule process (Steel & McKenzie, 2001), the Optimised Relaxed Clock model for molecular clocks (Douglas, Zhang & Bouckaert, 2021) for two datasets (two mitochondrial genes only and full set of four genes) were identified in BEAUti v.2.7.7 (Drummond et al., 2012), other priors were set up by default (Sarver et al., 2019). We used sequences from Pleuroxus aduncus as an outgroup. Estimation of the divergence time was performed based on a BI tree in BEAST2 for set of mitochondrial loci based on substitution rate of 0.7–1.8% per Myr (Colbourne & Hebert, 1996; Schwentner et al., 2013; Cornetti et al., 2019). In each analysis, we conducted four independent runs of MCMC (100 M generations, with selection of each 10 k generation). Effectiveness of MCMC was controlled by a universal analysis of log-files BEAST-2 in Tracer v.1.7.2 (Rambaut et al., 2018) to evaluate, do parameters reach a convergence based on ESS > 200. A consensus tree based on maximum MCC was constructed in TreeAnnotator v.2.7.7 (Drummond et al., 2012) with burn-in rate of 20%. To estimate the branch support values, we posterior probabilities (Nascimento, Reis & Yang, 2017). In addition to the molecular clock based on BI, a direct calculation of the divergence time for the major clades and subclades was performed based on a direct re-calculation of p-distances to time based on substitution rate of 0.7–1.8% per Myr (Colbourne & Hebert, 1996; Schwentner et al., 2013; Cornetti et al., 2019).

A haplotype network was constructed based on Neighbor-Joining Network algorithm (Bandelt, Forster & Rohl, 1999) in popART v.1.7 (Leigh, Bryant & Nakagawa, 2015). Such networks are regarded as better instruments for reconstructions of polytomies and unresolved topologies than the trees, representing phylogeographic relationships in space (Posada & Crandall, 2001). As an alternative to the Neighbor-Joining Network, we used Parsimony Splits network (Bandelt & Dress, 1992) in SplitsTree App v.6.3.27 (Huson & Bryant, 2006), minimizing the number of evolutionary steps necessary for interpretation of a data set and not repeating the MP tree with its topology, splitting the ambiguous topologies (Huson, Rupp & Scornavacca, 2010).

Results

BI and ML trees based on two mitochondrial genes were congruent in main clades, due to this we represent only the former, with marked branch support from both analyses. This tree represented in Fig. 3 clearly demonstrated that the genetic distances within the P. truncatus group are very small as compared to the distances between P. truncatus and the outgroup (P. aduncus). All totality of the sequences could be subdivided into two major clades, each with a relatively high SH-aLRT and high UFboot2 support: A–having a predominantly European distribution with a single exclusion (Ple025 from Western Siberia), and

B–having a predominantly Asian distribution, but also present in northern (095 and 096) and central (016, 097, 099 and 0121) portions of European Russia.

Figure 3 Maximum likelihood tree representing the diversity among phylogroups of Pleuroxus truncatus based on sequences of two mitochondrial genes (COI+12S).

Full ML tree (A) and the same with branches transformed to cladogram (B). The support values of individual nodes are based on SH-aLTR/bootstrap UFBoot2 tests (in percent).

The clade A is represented by few sequences, and its internal structure is not discussed here. Within the Major Clade B, there are three sub-clades having a moderate or low support: B1–from Western Siberia only;

B2–from Western Siberia and Mongolia;

B3–with a wide distribution, and just European populations also belong here.

Multilocus tree represented in Fig. 4 is similar to the mitochondrial tree in its topology, two same major clades (with a high support), and the same subclades (with a relatively high BI support for the sub-clades D1 and B2, but with a low support of the sub-clade B1) are well-recognizable.

Figure 4 Maximum likelihood tree representing the diversity among phylogroups of Pleuroxus truncatus based on sequences of four mitochondrial and nuclear genes (COI+12S+18S+28S).

Full ML tree (A) and the same with branches transformed to cladogram (B). The support values of individual nodes are based on SH-aLTR/bootstrap UFBoot2 tests (in percent).

Two major clades, A and B (see Table 2), are separated based on the TaxonomR analysis, although a locus representation was not complete in some cases. But, at the same time, we found that the population structure is complicated in Pleuroxus truncatus. A relatively high haplotypic and nucleotide diversity together with a low portion of G+C are characteristic of mitochondrial loci. Neutrality tests for different clades suggested similar demographic processes in them, and in different loci. For the overarching data compilation, values of Fs significantly less than zero and R2 approaching zero may indicate a substantial genetic differentiation within this species, as well as a likelihood of population decline and the occurrence of admixture processes.

Table 2 Genetic diversity of Pleuroxus truncatus group.

Groups	N	G+C	n	S	h	Hd	Pi	k	Fs	R2	CS	p (A/B)	
	COI (mitochondrion coding)	
Total COI	72	0.410	622	62	52	0.982	0.010	6.51	−4.91	0.048	g/a	0.0157	
Clade A	16	0.411	622	20	13	0.975	0.008	5.47	−5.28	0.117	a	0.0063	
Clade B	56	0.410	622	45	39	0.973	0.007	4.55	−3.52	0.045	g	0.0092	
	16S (mitochondrion rDNA)	
Total 16S	46	0.386	370	9	10	0.797	0.004	1.43	−3.66	0.078	g	0.0041	
Clade A	4	0.386	370	0	1	–	–	–	–	–	–	0.00	
Clade B	40	0.386	370	7	8	0.810	0.004	1.42	−1.98	0.101	g	0.004	
	18S (nuclear rDNA)	
Total 18S	49	0.547	629	24	19	0.890	0.005	2.88	−9.12	0.055	g/a	0.0062	
Clade A	7	0.545	629	1	2	0.476	0.001	0.47	0.59	0.238	–	0.0011	
Clade B	42	0.547	629	23	17	0.863	0.004	2.61	−8.42	0.052	g/a	0.0052	
	18S (nuclear rDNA)	
Total 28S	45	0.566	366	1	2	0.087	0.001	0.086	−0.79	0.043	–	0.0002	
Clade A	10	0.566	366	0	1	–	–	–	–	–	–	0.00	
Clade B	35	0.566	366	1	2	0.111	0.001	0.111	−0.572	0.055	–	0.00	
Note:

N, number of sequences; G+С, guanine-cytosine content; n, total number of sites (excluding sites with gaps/missing data); S, number of segregating (polymorphic) sites; Hd, haplotype diversity; h, number of haplotypes; Pi, nucleotide diversity per site; k, average number of nucleotide differences; Fs, Fu’s neutrality statistic (Fu, 1997); R2, Ramos-Onsins and Rozas R2-statistic (Ramos-Onsins & Rozas, 2002); P, “simple” p-distance between clade A and B (for “total” line) and within each group (for “clade” line); CS, coalescent simulation (Rozas et al., 2017) of demographic parameters (n, neutral; g, growth; d, decline; b, bottleneck; a, admixture).

Existence of two relatively independent major clades (“A” and “B”) was confirmed by AMOVA. At the first level (full dataset, absence of partitioning), analysis demonstrated significant differences within it (p < 0.001, rate on intra-populational variability = 42%). At the second level (two major clades from TaxonomR), the analysis confirmed a statistically significant separation of them (FST = 0.57, p < 0.05; the rest of variability is intra-populational one); at the third level (within two major phylogenetic lineages), the analysis demonstrated lacking of significant variation (a dispersion rate <0.5%; p > 0.05). The AMOVA results surely confirm the existence of two major phylogenetic lineages, plus some differentiation within the major clade B also takes place.

Coalescent simulation in dnaSP revealed that the most probable demographic mode of P. truncatus was population split or admixture, but a population growth was also possible within the major clade “B” (Table 2).

Neighbor-Joining network for COI haplotypes is represented in Fig. 5A. It is clearly subdivided into two portions well-corresponding to the major clades A and B.

Figure 5 COI haplotype networks of Pleuroxus truncatus.

Ones based on Neighbor-Joining algorithm, with nucleotide substitutions represented by crossing markers (A) and Parsimony Splits algorithm, with number of nucleotide substitutions proportional to branch lengths (B). Abbreviations: CeRus, central portion of European Russia; NWRus, North-West of European Russia and neighboring northern portion of Western Siberia; Povolzh, Povolzhye Region, a South-Central portion of European Russia belonging to the Volga River basin; WSib, Western Siberia; ESib, Eastern Siberia (without Yakutia Republic); Yakut, Yakutia=Sakha Republic; Mong, Mongolia.

The major clade A is represented by a limited number of sequences, and it is premature to discuss its inner structure. But it contains only European haplotypes, and interconnected to the major clade B through a single Western Siberian haplotype.

The major clade B has a shallow structure, with haplotypes subdivided mainly by 1–2 mutations; two star-like shapes could be found, but they are interconnected by two substitutions only. In the first star-shape, the central haplotype is distributed in Western Siberia and Yakutia. Two different Mongolian lineages, three Yakutian lineages, and a NW Russian lineage could be regarded as derived ones from the former. The central haplotype of the second star-shape is distributed in Western Siberia and the center of European Russia; one of its derived haplotypes in present in Povolzhye (Middle Volga basin). Haplotypes belonging to the sub-clades B1 and B2 occupy specific portions of the network.

The Parsimony Splits network suggests existence of two large clusters fully corresponding to the major clades A and B a demonstrating a story very similar to that in Fig. 5A. Cluster A contains mostly European members, with a single exception; central “haplotype” of the cluster B contains mainly Siberian members, with few exceptions, and several Siberian lineages, plus two Povolzhye lineages.

Estimations of the clade differentiation time was somewhat different based on different methods, but all such estimations for clades A-B, subclades B1–B2–B3 differentiation lie in the interval of ca. 0.01–0.3 Mya (Table S2). Therefore, the whole revealed pattern is very young, related to Late Pleistocene. Moreover, terminal clades of our tree (populations) are apparently of a Holocene age.

Discussion

In our dataset, both protein-coding and ribosomal loci were represented that allows us to accurately characterize real major phylogenetic lineages of Pleuroxus truncatus. Based on our data we can conclude that two major phylogenetic lineages exist within it. But we found that the phylogeographic structure of P. truncatus is very young (of Late Pleistocene-Holocene age) and shallow. It is consistent with a single widely distributed species, even subspecies could not be separated because the distribution ranges of two major clades are strongly overlapping. To date, we have no adequate information on South Europe; such populations need to be revised.

This pattern apparently is different from those of the Chydorus sphaericus group or Daphnia magna group, where “eastern” and “western” major phylogroups differ by many mutations, and the latter phylogroups could be recognized as subspecies or even independent species (Kotov et al., 2016; Bekker et al., 2018). Moreover, the southern half of Western Siberia is almost impenetrable for both “eastern” and “western” groups, because the water bodies here were rapidly colonized after the Late Pleistocene de-aridization (this area was not glaciated) by local endemic clades, surviving in the refugia of the pre-mountain areas of Altay and Sayan mountains and made a “life shield” for penetration of their relatives from east and west (Bekker et al., 2018; Zuykova et al., 2019). Such clades, namely earlier derived from the major clade B subclades B1 and B2 (supported by BI analysis, although having a moderate support in ML analysis), are found here also, and at the same geographic position (Fig. 2). But note that the differentiation within the major phylogroup B is not so obvious as in Daphnia: even the topology of subclades B2 and B3 in Fig. 3 (2-gene tree) and Fig. 4 (4-gene tree) are not the same, but the Western Siberian sub-clade B1 is well-supported in all our analyses confirming applicability of the “life shield” hypothesis to the case of P. truncatus.

We have found some traces of such a pattern in P. truncatus: two longitudinally differentiated major clades seen in both trees and haplotype networks, earlier derived clades survived in Western Siberia (Fig. 2), etc. Both population split or admixture and population growth were demographic models explaining this taxon pattern; population decline and admixture processes were common in its evolutionary history (Table 2). This pattern is more shallow, probably, because it is younger than the pattern of the cladocerans described above (Garibian et al., 2021). It seems that the total population of P. trncatus was not so strongly split by the unfavorable conditions during the Pleistocene cold phases.

Note that just COI haplotype networks were used several times for the reconstruction of the cladoceran dispersion history in Late Pleistocene-Holocene. We can conclude that the mutation accumulation rate was lower in littoral P. truncatus, and is a main explanation of aforementioned differences between its phylogeographic pattern as compared to that in planktonic Daphnia and Moina and eurytopic Chydorus (Bekker et al., 2016; Kotov et al., 2016; Bekker et al., 2018). Neustonic Scapholeberis and Megafenestra have an even higher mutation rate and higher levels of divergence as compared to non-neustonic daphniids, with very well differentiated biological species (Taylor, Connelly & Kotov, 2020). We can hypothesize that a low mutation rate in Pleuroxus could be explained partly by a lower level of UV radiation in the darker vegetation zone as compared to open water plankton. A general comparison of the microevolution speed in planktonic and littoral animals in the Cladocera, as well as its dependence on different dispersion strategies sensu Fryer (Fryer, 1972), could be interesting topic of further studies, but to date any speculations based on few investigated representatives are not convincing. Note that great differences in the mutation rates between different taxa were demonstrated previously in other freshwater animals, e.g., molluscs (Bolotov et al., 2016, 2017).

A great challenge is now established for cladoceran phylogeography: invasive species (Kotov, Karabanov & Van Damme, 2022). They were found in almost each (!) previous phylogeographic study by our team (see Bekker et al., 2018; Karabanov et al., 2018; Garibian et al., 2020; Neretina et al., 2021). We cannot fully exclude a chance of non-indigenous haplotypes in our datasets originating due to human-mediated biological invasions. Such haplotypes introduce a significant uncertainty to population models (Karabanov, Bekker & Kotov, 2020) and lead to erroneous phylogeographic conclusions (Karabanov et al., 2022a). Appearance of the clade B in Povolzhie and the northern portion of European Russia was, most probably, related to a series of natural dispersions from Siberia. As a mechanism, we can regard ephippium transportation by migrating water fowl being a main dispersal factor for the invertebrates in the pre-anthropogenic times (Viana, Santamaria & Figuerola, 2016; Coughlan et al., 2017; Green et al., 2023). But presence of the central haplotype of the second star-shape (Fig. 5A) just in the Botanical Garden in Moscow could be regarded as an unknown case of P. truncatus cryptic invasion.

But even such assumption does not change our main conclusion: P. truncatus has a phylogeographic pattern shallower and younger relative to previously studied taxa.

Conclusions

Our pioneer study of a phylogeographic pattern in a “non-Chydorus” chydorid based on a material from a high range in North Eurasia revealed a clear differentiation between western and eastern major mitochondrial phylogroups. But we found that the phylogeographic structure of P. truncatus is very young (of Late Pleistocene-Holocene age) and shallow. Our investigation filled obvious gaps in our knowledge on the non-planktonic cladoceran phylogeography and evolutionary history and revealed similarities of Pleuroxus truncatus phylogeographic patterns with, and differences from, those previously described in Chydorus and Daphnia.

Supplemental Information

Supplemental Information 1 Sequences used in this study.

Includes information on sampled localities.

Supplemental Information 2 Major clade and subclade differentiation time based on direct re-calculation of mutation rate and molecular clocks based on mitochondrial gene tree and the tree based on all gene datasets.

Many thanks to R. J. Shiel for linguistic corrections in an earlier draft. All SEM works were carried out at the Joint Usage Center “Instrumental Methods in Ecology”, A. N. Severtsov Institute of Ecology and Evolution of Russian Academy of Sciences.

Additional Information and Declarations

Competing Interests

The authors declare that they have no competing interests.

Author Contributions

Alexey A. Kotov conceived and designed the experiments, prepared figures and/or tables, authored or reviewed drafts of the article, and approved the final draft.

Petr G. Garibian performed the experiments, prepared figures and/or tables, authored or reviewed drafts of the article, pre-sorting of specimens and their identification, and approved the final draft.

Anna N. Neretina performed the experiments, analyzed the data, authored or reviewed drafts of the article, and approved the final draft.

Dmitry P. Karabanov conceived and designed the experiments, performed the experiments, analyzed the data, prepared figures and/or tables, authored or reviewed drafts of the article, and approved the final draft.

DNA Deposition

The following information was supplied regarding the deposition of DNA sequences:

The DNA sequences are available at NCBI GenBank: PQ798952–PQ799024 for COI, PQ799025–PQ799071 for 16S, PQ799072–PQ799121 for 18S and PQ799124–PQ799169 for 28S.

Data Availability

The following information was supplied regarding data availability:

The raw data is available in the Supplemental File.

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
