# Peer review of "A phylogeographic pattern of the trans-Palaearctic littoral water flea Pleuroхus truncatus (O.F. Müller, 1785) (Cladocera: Chydoridae)"

_PeerJ, doi:10.7717/peerj.19355_

## Round 0.1 · original submission · Major Revisions

Three reviewers have completed their review of the manuscript. They have given expert comments from different perspectives and, as you can see, you should pay more attention and respond to the diverse concerns of three reviewers, especially reviewer 2, on a case-by-case basis.

·

Basic reporting

The study explores the phylogeographic patterns of the littoral water flea, Pleuroxus truncatus, across the trans-Palaearctic region, providing valuable insights into the evolutionary history, genetic diversity, and geographic distribution of this species.
The paper offers new and significant contributions to our understanding of the phylogeography of littoral Cladocera, a group for which phylogeographic patterns are poorly understood. In contrast, planktonic cladocerans, particularly Daphnia, have well-established and extensively studied phylogeographic patterns. This highlights the importance of the study in filling a significant gap in our knowledge of littoral taxa.

Minor suggestions:
The aims of the study are presented in just one sentence ("The aim of our paper is to study the phylogeographic pattern of P. truncatus in northern Eurasia"), which closely mirrors the title. The authors might consider expanding and enhancing the aims to better reflect the broader contributions of their work. For example, they could emphasize how the study provides new insights into the phylogeography of littoral Cladocera, incorporates molecular clock analyses, and conducts a biogeographic reconstruction, which is well-described and thoroughly discussed in the manuscript.

Please add information about scanning electron microscope (Figure 1) to Methods.

The authors should also carefully review the text (e.g. Line 21: the genus name Daphnia should be italicized; Line 44: there is a missing space that needs correction) and the construction of some sentences to improve clarity and readability throughout the manuscript. While the paper is strong overall, refining the language and sentence structure would further enhance its quality and presentation.

Experimental design

no comment

Validity of the findings

no comment

Additional comments

no comment

Reviewer 2 ·

Basic reporting

Linguistic issues scattered throughout the text obscure the meaning. Some examples are provided below:

Currently among North Eurasian cladocerans, representatives of a single anomopod family, Chydoridae Dybowski et Grochowski, 1894, comprise 40% of the total species number (113 of 287 taxa (Korovchinsky et al., 2021)). The same proportion is characteristic of the world fauna (e.g. 269 among 620 species according to (Forro et al., 2008), although such numbers are now outdated).

Apparently, phylogeographic patterns of the aforementioned well studied planktonic animals could be different from those of littoral taxa, e.g. due to different methods of resting egg dispersion: the latter are well-adapted in planktonic forms to be easily dispersed, but in littoral forms frequently are adapted to be kept in the water body where they were produced (Fryer, 1972).

Analysis in TaxonomR led in clearly subdivision of all our genetic dataset into two Major Clades, A and B (see Table 2), although a locus representation was not complete in some cases.

The authors should revise their language to avoid any ambiguous expressions.

Experimental design

no comment

Validity of the findings

1. Tree structures in Fig. 3 and Fig. 4 are very hard to read and the support values were not provided on most of the nodes, even on some key nodes of the proposed Major Clades and subclades.

2. Figure legend of Fig. 5 is not clear, impeding the interpretation of results, especially in comparison with the clades from phylogenetic trees. Legend of Fig.1 should also include explanations of each structure showed in A-F.

3. Conclusions did not effectively reflect the main findings of the research.

Annotated reviews are not available for download in order to protect the identity of reviewers who chose to remain anonymous.

Reviewer 3 ·

Basic reporting

Phylogeography research is relevant and in demand among zoologists and ecologists. The article is written in clear scientific English. The introduction and the list of references reflect the current research on the subject. The article contains high quality and relevant figures. The conclusions fully correspond to the stated objectives of the article and emphasize the importance of such research.

Experimental design

The study is in line with the aims and themes of the journal and fills a gap in the knowledge of the phylogeography of one of the most widely distributed species of the genus Pleuroxus Baird, 1843 (Crustacea: Anomopoda: Chydoridae). The study was carried out in accordance with high technical and ethical standards. The methods used in the work are described with sufficient detail & information to replicate.

Validity of the findings

The validity of the findings is not in doubt.

Additional comments

In the “Field collection and provisory identification” section, the make and model of microscopes used by the authors should be added.

Information on specific sample collection sites should be added to the “Field collection and provisory identification” section. Names of water bodies or areas where samples were collected.

Were samples from the Zeya River basin used? The studies of P.G. Garibian indicate that Pleuroxus truncatus is found in this region.

---

## Round 0.2 · Minor Revisions

Thank you for your submission to PeerJ.

Please change your work as reviewers' comments.

·

Basic reporting

The authors effectively combined molecular data with biogeographical analyses, providing valuable insights into the dispersal history of this species. In the second round of review, all my previous comments have been fully addressed. Additionally, the manuscript's language has been improved, enhancing its readability and coherence.

Experimental design

The manuscript is well-structured, and the results are presented clearly and convincingly. The conclusions are well-supported by the analyses and appropriately justified. In my opinion, the current version of the article meets the standards of PeerJ and is suitable for publication in its present form.

Validity of the findings

-

Reviewer 2 ·

Basic reporting

The manuscript has been improved and the main issues from the previous version including language and figure readability issues have been addressed.

I have a few minor comments:

line 60-61: "only few species are able to penetrate plankton." I don't quite understand what this means
line 66: "colud" should be "could"
line 232: "To estimate the branch support values, we posterior probabilities". A verb is missing.
line 233-234: "BI and ML trees were congruent in main clades, due to this we represent only the former, with marked branch support from both analyses." This sentence should be move to the Result section.
line 276: "a population structure" should be "the population structure"
line 290: "в" should be deleted
line 345: "althouth" should be "although"
line 391: "phylogeograpic" should be "phylogeographic"

Experimental design

no comment

Validity of the findings

The main findings from different analyses (trees, haplotype, AMOVA, etc.) supported the differentiation between clades A and B, which is sound and provides important understanding on the phylogeography of P. truncatus. Subclade B1 is also well-supported. However, it seems that the topology of subclades B2 and B3 in Fig. 3 (2-gene tree) and Fig. 4 (4-gene tree) are not the same. In the 2-gene tree, subclade B2 includes 6 individuals: 018, 086, 084, 081, 082, 088. But in the 4-gene tree, subclade B2 includes 4 different individuals: 090, 089, 091, 092. Also, in the 4-gene tree, subclade B3 is not monophyletic, thus is not a "clade". The author should consider whether they need to differentiate B2-B3 as subgroups. Merging them together seems not affect the main conclusions of the paper.


The Conclusions is revised to only one sentence, which is quite general and not informative. I would suggest the authors to summarise the main findings and highlights of the work in the conclusion, especially the amount of materials you're able to examined, the clear differentiation between clades A and B and its phylogeographic implications, and the gap in the previous study of this group and how your work contributes to the understanding of its phylogeography and evolution.

Additional comments

no comment

---

## Round 0.3 · accepted · Accept

Congratulations!

Thank you for submitting your work to PeerJ.

Reviewer 2 ·

Basic reporting

All the previous comments have been addressed and I have no more revisions to the manuscript.

Experimental design

-

Validity of the findings

-

Additional comments

-